# Evolved Policy Gradients

**Rein Houthooft**
OpenAI

**Richard Y. Chen**
OpenAI

**Phillip Isola**
OpenAI

**Bradly C. Stadie**
UC Berkeley

**Filip Wolski**
OpenAI

**Jonathan Ho**
OpenAI

**Pieter Abbeel**
UC Berkeley

## Abstract

We propose a meta-learning approach for learning gradient-based reinforcement learning (RL) algorithms. The idea is to evolve a differentiable loss function, such that an agent, which optimizes its policy to minimize this loss, will achieve high rewards. The loss, parametrized via temporal convolutions over the agent's experience, enables fast task learning and eliminates the need for reward shaping at test time. Empirical results show that our evolved policy gradient algorithm achieves faster learning on several randomized environments compared to an off-the-shelf policy gradient method.

## 1 Introduction

When a human learns to solve a new control task, such as playing the violin, they don't require external rewards to start learning. They immediately have a feel for what to try, and for whether or not they are making progress towards the goal. Effectively, humans have access to a very well shaped internal reward function, derived from prior experience. Our aim in this paper is to devise reinforcement learning (RL) agents that similarly have a prior notion of what constitutes making progress on a novel task. Rather than encoding this knowledge explicitly through memorized behaviors, we encode it implicitly through a learned loss function.

Our method consists of two optimization loops. In the inner loop, an agent learns to solve a task, sampled from a task distribution, by minimizing a loss function provided by the outer loop. In the outer loop, the parameters of the loss function are adjusted so as to maximize the final returns achieved after learning by inner loop agents. The inner loop is optimized via stochastic gradient descent (SGD) while the outer loop is optimized via evolution strategies (ES) (2; 5; 1; 3). Figure 1 provides a high-level overview of this approach.

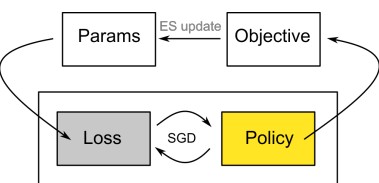

Figure 1: High-level overview of our approach. See text for details.

## 2 Methodology

We assume access to a distribution $p(\mathcal{M})$ over Markov decision processes (MDPs). Given a sampled MDP $\mathcal{M}$, the inner loop optimization problem is to minimize the loss $L_\phi$ with respect to the agent's policy $\pi_\theta$:

$$\theta^* = \arg\min_\theta \mathbb{E}_{\tau \sim \mathcal{M}, \pi_\theta}[L_\phi(\pi_\theta, \tau)]. \tag{1}$$

The outer loop objective is to learn $L_\phi$ such that an agent policy $\pi_{\theta^*}$ trained with the loss function actually does achieve high expected returns, $R$, in the MDP distribution:

$$\phi^* = \arg\max_\phi \mathbb{E}_{\mathcal{M} \sim p(\mathcal{M})} \mathbb{E}_{\tau \sim \mathcal{M}, \pi_{\theta^*}}[R_\tau]. \tag{2}$$

---

**Algorithm 1:** Evolved Policy Gradients (EPG)

---

1  **Input:** Learning rates $\delta$ and $\alpha$, noise standard deviation $\sigma$, environment distribution $p(\mathcal{M})$
2  **[Outer Loop] for** epoch $e = 1, \ldots, E$ **do**
3     **[Inner Loop] for** each worker $i = 1, \ldots, n$ **do**
4         Sample random vector $\epsilon_i \sim \mathcal{N}(\mathbf{0}, \mathrm{I})$ and calculate the loss function parameter $\phi + \sigma\epsilon_i$
5         Generate a random environment $\mathcal{M}_i$ according to $p(\mathcal{M})$
6         **for** trajectory $j = 1, \ldots, K$ **do**
7             Sample initial state $s_0$
8             **for** timestep $t = 0, \ldots, M$ **do**
9                 Sample action $a_t$ from $\pi_{\boldsymbol{\theta}}(a_t|s_t)$
10                 Take action $a_t$, observe reward $r_t$ and next state $s_{t+1}$ from $\mathcal{M}_i$
11                 If termination signal is reached, reset environment, resampling initial state $s_0$
12             Update policy parameter $\boldsymbol{\theta}$ based on the loss function $L_{\phi+\sigma\epsilon_i}$ according to Eq. (3)
13         Compute the final return $R_i$
14     Update the parameter $\phi$ for the loss function $L_\phi$ according to Eq. (4)

---

## 2.1   ALGORITHM

The final episodic return $R_\tau$ at evaluation cannot be represented as an explicit function of the loss function $L_\phi$, and thus we cannot use gradient-based methods to directly solve Equation (2). Our approach, summarized in Algorithm 1, relies on evolution strategies to optimize the loss function in the outer loop.

At the start of each iteration in the outer loop, we generate a standard multivariate normal vector $\epsilon_i \in \mathcal{N}(0, \mathrm{I})$ with the same dimension as the loss function parameter $\phi$ for each inner loop worker $i \in \{1, \ldots, n\}$. The outer loop gives each inner loop worker a perturbed loss function

$$L_i = L_{\phi+\sigma*\epsilon_i},$$

with perturbed parameters $\phi + \sigma * \epsilon_i$ where $\sigma$ is the standard deviation.

Given a loss functions $L_i$ from the outer loop, each inner loop worker $i$ samples a random MDP from the task distribution, $\mathcal{M}_i \sim p(\mathcal{M})$. The worker then trains a policy $\pi_{\boldsymbol{\theta}}$ in $\mathcal{M}_i$ over $K$ trajectories $\{\tau_j\}_{j=1}^K$ of $M$ timesteps of experience. After each trajectory, the policy takes a gradient step with respect to minimizing the loss function $L_i$:

$$\boldsymbol{\theta} \leftarrow \boldsymbol{\theta} - \delta \cdot \nabla_{\boldsymbol{\theta}} L_i(\pi_{\boldsymbol{\theta}}, \tau_j). \qquad (3)$$

At the end of the inner-loop training, each worker returns the final return $R_i$[1] to the outer loop. The outer-loop aggregates the final returns $\{R_i\}_{i=1}^n$ from all workers and updates the loss function parameter $\phi$ as follows:

$$\phi \leftarrow \phi + \alpha \cdot \frac{1}{n\sigma} \sum_{i=1}^n R_i\epsilon_i, \qquad (4)$$

where $\alpha$ is the outer loop learning rate.

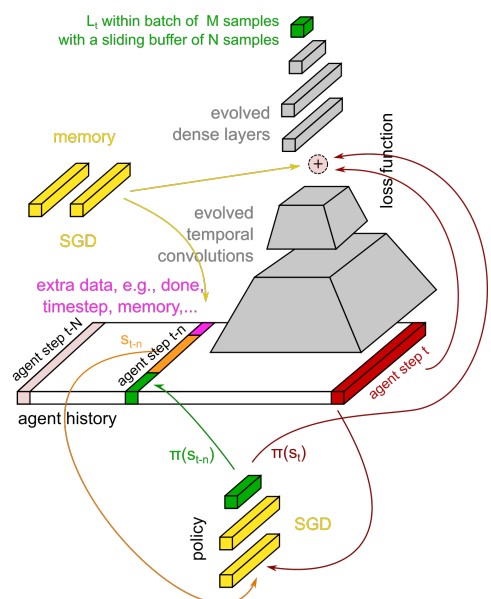

Figure 2: Architecture of a loss computed for timestep $t$ within a batch of $M$ sequential samples, using temporal convolutions over a buffer of size $N$: dense net on bottom is the policy $\pi(s)$, taking as input the observations (orange), while outputting action probabilities (green). Green block on the top represents loss output. Gray blocks are evolved, yellow blocks are updated through SGD.

---

[1] More specifically, $R_i = R_{i,K} + \frac{1}{K} \sum_{t=0}^K \frac{(t+1)}{K} R_{i,t}$ with $R_{i,j}$ the undiscounted return of episode $t$ in worker $i$.

## 2.2 ARCHITECTURE

The agent is parametrized using an MLP policy. The agent has a memory unit to assist learning in the inner loop. The memory parameters are updated during training in the inner loop but does not directly impact the agent's policy output. Instead, the memory is feed as an input to the loss, which modulates the learning process. An experience buffer stores the agent's $N$ most recent timesteps.

The loss function does not observe the environment rewards directly. However, in cases where the reward cannot be fully inferred from the environment, such as the forward-backward random Hopper in Section 3, we augment the inputs to the loss function with reward.

In practice to bootstrap the learning process, we initially add to $L(\phi)$ a guidance PPO (4) surrogate loss function, making the total loss

$$\hat{L}_\phi = (1 - \alpha)L_\phi + \alpha L_{\text{PPO}},$$

with $\alpha$ annealed from 1 to 0, so that by the end of training $\hat{L}_\phi = L_\phi$.

## 3 EXPERIMENTS

We apply our method to two randomized MuJoCo environments, namely Hopper (with randomized gravity, friction, body mass, and link thickness) and Reacher (with randomized link lengths). We compare learning performance using EPG versus PPO (4). Figure 3 shows learning curves for these two methods on the randomized Hopper and Reacher environments.

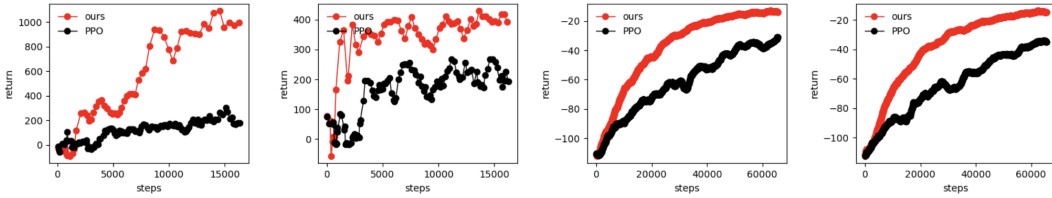

Figure 3: Learning curves for PPO (black) vs no-reward EPG algorithm (red). Left two subplots are two different randomizations of Random Hopper, right two are two different randomizations for Random Reacher.

In both cases, the PPO agent observes reward signals whereas the EPG agent does not observe rewards. Nonetheless, the EPG agent learns more quickly and obtains higher returns compared to the PPO agent. This indicates that our method generates an objective that is more effective at training agents, within these task distributions, than PPO. This is true even though the learned loss does not observe rewards at test time. This demonstrates the potential to use our method when rewards are only available at training time.

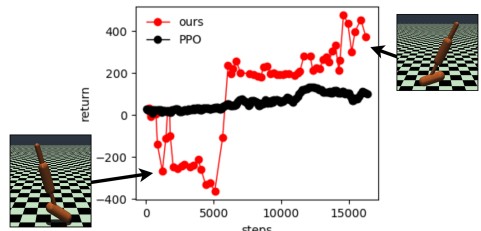

Figure 4: Exploratory behavior of EPG (red) in forward-backward random Hopper in comparison to PPO (black). The EPG agent goes backwards for a while, then going forward and exploiting. Inlays indicate qualitative behavior observed in these two phases.

To understand whether EPG trains agents that explore, we test EPG and PPO on a specialized forward-backward random Hopper environment, with random reward for either forward or backward hopping. Note that without observing the reward, the agent cannot infer whether the Hopper environment desires forward or backward hopping. Thus we provide the environment reward to the loss function in this setting. Figure 4 shows the learning curves of a PPO agent and an EPG agent, and shows that EPG manages to explore both forward and backward hopping whereas PPO does not.

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
