# OpenReview forum: "Evolved Policy Gradients"
_ICLR.cc/2018/Workshop — Reject_

### Official Review · AnonReviewer3 · 2018-03-04
**Approach lacks sufficient motivation, justification, and explanation**

**Rating:** 4
**Confidence:** 4

**Review:**

This paper proposes to evolve a differentiable loss function in gradient-based reinforcement learning algorithms. Specifically, the authors consider a scenario where the learner is able to choose among a parametrized family of loss functions, and they propose an epoch-based algorithm where the learner optimizes the agent's policy according to a specific loss function within each epoch and updates the loss function between epochs.

The paper presents an interesting approach for gradient-based reinforcement learning algorithms. The general algorithm seems for the most part natural and broad enough in scope to be of interest most of the reinforcement learning community.

At the same time, I found many important details of the presentation to be unclear and the major results to be insufficiently motivated. In particular, the authors do not explain why evolving a loss function may be a better approach than using environment rewards. Moreover, the parametric form of the loss function seems like a crucial aspect of this type of framework, and yet it is never discussed, even in the experiments. Moreover, the authors also do not touch upon evolution of the loss function and/or the policy gradients. The final results on their own are not very convincing or informative about how and why the method works.

Specific comments and questions:
1) tau not defined on in equation (1) and onwards. It's presumably a trajectory?
2) "The final episodic return R\tau at evaluation cannot be represented as an explicit function of the loss function L_\phi". I'm not sure why this is true if L_\phi takes \tau as input.
3)  Equation (4) isn't motivated.
4) The form of the R_i is presented without motivation.
5) "The loss function does not observe the environment rewards directly." Why? It seems like they are available, and they are in fact used later on in the forward-backward Hopper environment.
5) The precise form of L_\phi isn't presented in the experiments.
6) There should at least be a sentence or two discussing related work and the novelty of this paper. For instance, [Reward Design via Online Gradient Ascent - Sorg, Lewis, and Singh] seems to present a similar idea with much better justification.

Minor comments:
1) Given a loss functions -> Given a loss function

Overall, I found this paper to be unclear in its novelty and significance and lacking in its motivation and justification.

---

### Official Review · AnonReviewer1 · 2018-03-10
**An interesting idea with some promising results - lacking some clarity/context but you can forgive this in the workshop setting.**

**Rating:** 7
**Confidence:** 4

**Review:**

Evolved Policy Gradients
=====================

This paper proposes a meta-learning approach that "evolves"="Evolution Strategies" for shaping rewards to be used with a policy gradient algorithm.
This algorithm operates over a distribution of environments and attempts to learn shaping rewards that are good across the distribution.

There are several things to like about the paper:
- The idea is appealing, with good motivation / intuition.
- The practicalities of how to implement this algorithm are well explained.
- There are experimental results that demonstrate a clear benefit versus standard policy gradient baselines.

However, there are also several shortcomings:
- The connection/relation to other evolutionary/metaRL strategies is not well-motivated or explained... how should we think of this versus MAML, versus RL^2, versus... the authors are no doubt aware of this work but we don't see a discussion or comparison.
- The connection of "evolutionary strategies" to policy gradient should be made more clear... at some level ES just performs policy gradient without gradients (finite difference)... could this whole algorithm then be simplified as just "do policy gradient on a meta-level"? If not, then why not?
- Some of the writing is rushed and confusing:
"The loss function does not observe the environment rewards directly. However, in cases where the reward cannot be fully inferred from the environment, such as the forward-backward random Hopper in Section 3, we augment the inputs to the loss function with reward."
Similarly "In practice to bootstrap the learning process"... why is "in practice" different to what has been described up to this point?

Overall I think the paper will be valuable to the ICLR workshop so I vote accept.

---

### Official Review · AnonReviewer2 · 2018-03-11
**Trying to overlook the actual submission**

**Rating:** 6
**Confidence:** 4

**Review:**

Before jumping in, if this were a review only of the submitted paper it would lean heavily towards rejection. There are glaring omissions of crucial details, unclear writing, and a complete lack of any sort of discussion of related work or effort to put this work into a context. Even considering page limitations, it is hard to take anything useful away from the actual submission. However, these complaints are addressed by the version on Arxiv, and so instead of wasting time nitpicking on things already fixed, consider this a review for the Arxiv version with an assumption that you could pair-down that version to something of similar quality within page limits.

The authors use a simple randomized hillclimbing algorithm, evolutionary strategies (ES) previously applied to policy optimization, to optimize the parameters of a convolutional network computing a loss function used to train a reinforcement learning agent by differentiating through the loss into the agent's policy (one of the inputs to the network). After optimizing the loss parameters, an agent minimizing it out-performs an agent using PPO. The method is demonstrated on two randomized control tasks, Hopper and Reacher, in which parameters of the environment are sampled randomly for each trial (not each episode).

It is important to emphasize that, during evaluation, the agent trained using the evolved loss does not see the true rewards, whereas the PPO agent does. This means that, for all the emphasis on 'loss functions' this is quite possibly better explained by reward optimization, which would also suggest reasons for not widening the domain distribution.

The comparison with PPO cannot truly serve as an evaluation of the method because PPO is being trained on one task and this larger system is being trained on many (although I am unsure of how 3000 epochs compares with the test-time training). If the argument was that the learned loss function is general across RL domains, then the comparison would be fair, but obviously the domains used would need to be expanded with only a single loss learned for all. Along these lines, does the Hopper loss work for the Reacher or vice versa? If you merge the two, so that each instance could come from either, does learning a loss for 'both' work? These, I think, would be much more interesting.

Although a bit incremental there is a definite insight to be drawn from this work, as well as some novelty in the exact form the meta-learning takes in this instance. The chief lesson I would take from this is that we can exchange human thought for parallel computation, and vice versa. The method is extremely simple, not even evolutionary in any meaningful way, and is able to produce a method that learns significantly faster than PPO. It may also hint at the promise of future combinations of evolutionary methods and RL. As mentioned, I think more understanding into how well the loss generalizes across domains would be very useful here. The effective number of possible Hopper and Reacher variations is likely not *that* large, so depending on the amount of training this could have overfit to them pretty heavily.

Pros: Simple, makes an interesting point. Combining 'evolutionary' methods and RL together is a promising direction, and this does so effectively without complicating either optimization.

Cons: Reading the submitted version was a waste of time, only consider reading the Arxiv version. Various small hacks included to make it work ('final returns' not final returns, annealing in the PPO loss with the optimized loss). Not clear on the amount of training / compute used, or the dimensionality of the parameters optimized, which we know to be a fairly important interaction with success of these types of methods.

---

### Decision · Program_Chairs · 2018-03-20
**ICLR 2018 Workshop Acceptance Decision**

**Decision:**

Reject

**Comment:**

This paper has not been accepted for presentation at the ICLR workshop. While the full version of the paper is interesting, out of fairness we can only consider the 3 page versions, which for this submission was not clear enough. However, the conversation and updates can continue to appear here on OpenReview.